# Capability of the TrueColor Sensor Array for Determining the Nitrogen Supply in Winter Barley (*Hordeum vulgare* L.)

**DOI:** 10.3390/s22166032

**Published:** 2022-08-12

**Authors:** Andreas Christ, Oliver Schmittmann, Peter Schulze Lammers

**Affiliations:** Institute of Agricultural Engineering, University of Bonn, Nußallee 5, 53115 Bonn, Germany

**Keywords:** CIELab color space, site-specific nitrogen fertilization, crop nutrient demand, crop management, SPAD meter, nitrogen dosing test

## Abstract

In agriculture, efforts are being made to reduce pesticides and fertilizers because of the possible negative environmental impacts, high costs, political requirements, and declining social acceptance. With precision farming, significant savings can be achieved by the site-specific application of fertilizers. In contrast to currently available single sensors and camera-based systems, arrays or line sensors provide a suitable spatial resolution without requiring complex signal processing and promise significant potential regarding price and precision. Such systems comprise a cost-effective and compact unit that can be extended to any working width by cascading into arrays. In this study, experiments were performed to evaluate the applicability of a TrueColor sensor array in monitoring the nitrogen supply of winter barley during its growth. This sensor is based on recording the reflectance values in various channels of the CIELab color space: luminosity, green–red, and blue–yellow. The unique selling point of this sensor is the detection of luminosity because only the CIELab color space provides this opportunity. Strong correlations were found between the different reflection channels and the nitrogen level (R² = 0.959), plant coverage (R² = 0.907), and fresh mass yield (R² = 0.866). The fast signal processing allows this sensor to meet stringent demands for the operating speed, spatial resolution, and price structure.

## 1. Introduction

Nitrogen from crop fertilizer that is not absorbed by plants has negative effects on the environment, such as groundwater pollution [1]. However, nitrogen deficiency can affect plant health and result in lower yields. Inadequate fertilization can be reduced by combining new technologies with farming practices to adapt to specific needs [2]. For more than two decades, digitized solutions have been introduced for site-specific field operations to save resources and protect the environment [3]. Sensor data can be applied to crop management to improve the yield of agricultural products, which has economic benefits. The challenge for sensor-based applications is to develop valid algorithms for the relationships between the sensor data and crop yield. The indices developed so far for sensor-controlled fertilization are based on computing spectral channels in the visible and near-infrared (IR) wavelength range [4]. Spectral analysis is based on the reflection behavior of plant cell compartments. The pigments within chloroplasts are divided into two photosystems: I (absorption maximum at a wavelength λ = 700 nm) and II (absorption maximum at λ = 680 nm) [5]. A reflection over the entire visual range of electromagnetic radiation appears white to the human eye and can be used for the white balance when calibrating optical reflection sensors. The absorption properties of pigments in the blue and red wavelength range produce the complementary color green, which results in the human perception of leaf color [6].

Various modeling approaches for spectral analysis are available in the literature. A distinguishing feature is the number of frequency bands. One approach to the development of indices is the multivariate calculation of up to three ranges of the reflection spectrum. Although healthy vegetation has a low reflection in the wavelength range of λ = 600–700 nm (red visible light), its reflection at λ = 700–1300 nm (near-IR light) is much greater [7]. A cost-effective approach to estimating the nitrogen content according to reflection ratio R_ratio_ (R_870_/R_620_) achieved coefficients of determination (R²) of 0.70 [8] and 0.82 [9]. Quality indices for determining the nitrogen content depend on both the crop type and growth conditions at the test sites [10,11,12]. Sufficient quality has been achieved by using different discrete wavelengths [13] and adapting mathematical pre-factors [10,14]. The partial least squares method is commonly used for the detection of herbal ingredients [15,16,17]. In contrast to approaches using index calculations, this type of multivariate regression requires pretreatment of the datasets. The signal noise is reduced by differentiating the raw signal; the second derivative eliminates outliers, which results in the highest R² values for nitrogen prediction [18].

In addition to the reflection spectrum of the plant cells themselves, both the fresh mass and degree of coverage influence the assessment of the plant condition. Online measurements of the fresh mass have shown strong correlations with the plant condition and thus the yield [19]. An independent and early yield assessment adapted the cultivation strategy [20,21]. A prerequisite for increasing the yield by nutrient supply is the yield potential of the soil. In case the soil is the limiting factor for plant growth, the specifics of the nutrient supply should be determined [22]. In this context, the main focus is on the nutrient absorption capacity of the plants [23]. The basis for the nutrient supply of crops such as cereals is the agronomic relationship between nitrogen uptake and yield uptake [24]. High grain yields depend on the number of ear-bearing stems, grain/ear ratio, and thousand kernel weight. To increase the number of grains per ear, applying fertilizer at the beginning of longitudinal growth is necessary. Grain filling is strongly influenced by nitrogen fertilization after the end of ear pushing. Meanwhile, the degree of plant cover is a good reflection of the number of ear-bearing stems and thus the stocking.

Statistical analysis has found strong correlations between the degree of coverage and the biomass in the red wavelength range (λ = 678 nm with a correlation coefficient r = −0.724) and near-IR range (λ = 721–1050 nm with r = 0.68) [25]. Studies have shown a correlation between the biomass and yield of up to R² = 0.56 [26]. This leads to the hypothesis that present sensors meet the requirements for accurately assessing plant coverage and facilitating early fertilization. Four different models for sensor-controlled nitrogen fertilization are currently available commercially. Across all manufacturers, the measurement spot size covers less than 20% of the working width: 17% for the N-Sensor ALS (Yara International ASA, Oslo, Norway), 11% for NEXT GreenSeeker (Trimble Inc., Sunnyvale, CA, USA), 5.5% for ISARIA PRO Active (Fritzmeier Umwelttechnik GmbH & Co. KG, Aying, Germany), and 5% for OptRx (Ag Leader Technology, Ames, IA, USA) [27]. However, with the TrueColor sensor array, coverage of 100% is possible. Using the implemented optical elements in broad segments of industry induces low unit costs and, therefore, cost-effective production of the sensor array.

This study mainly aimed to evaluate the TrueColor sensor array as a low-cost approach to determining the nitrogen supply for crops. The field performance of this sensor array was analyzed in terms of the following aspects:Determination of the required detection area;Influence of the plant coverage on the measurement quality;Comparison of the correlations of the reflection channels to the agronomic parameters and SPAD value.

## 2. Materials and Methods

### 2.1. Site Description

Experiments were conducted on the experimental farm Campus Klein-Altendorf (50°37′51″ N, 6°59′32″ E) of the University of Bonn. The soil type was luvisol and comprised mostly loamy silt or clayey loam. At this site, the annual precipitation is 603 mm, the annual average temperature is 9.4 °C, and the average vegetation period is 165–170 days [28,29].

As a representative crop for Nitrogen indication, winter barley (Quadriga variety) from the seed breeding company Secobra Saatzucht GmbH (Lemgo, Germany) was used. The multi-row variety is mainly marketed as animal feed. After plowing, the winter barley was sown on 28 September 2019. Herbicides, fungicides, insecticides, and growth regulators were applied uniformly according to standard operating practice at the experimental farm. Basic fertilization of the macronutrients phosphorus and potassium occurred in late summer 2019 with granulated mineral fertilizers. All further fertilization was done with ammonium nitrate urea solution in liquid form with field sprayers and drag hoses. In February 2020, soil samples were taken at a depth of 90 cm to determine the fertilizer requirements, and the mineralized nitrogen content (N_min_) was determined for 30 cm-thick layers. The top (0–30 cm), middle (30–60 cm), and deepest (60–90 cm) layers had N_min_ = 3, 4, and 2 kg N ha^−1^, respectively. In total, 9 kg N ha^−1^ was considered to be available to plants for the calculation of the fertilizer demand.

### 2.2. TrueColor Sensor Array

The measurement principle of the TrueColor sensor is based on recording the reflected radiation on a true color scale [30]. The sensor was developed in collaboration with Premosys GmbH (Kalenborn-Scheuern, Germany) and was designed as an array with the type designation PR0262. 

Five individual sensors, each with its own light source and a segment width of 10 cm, are arranged next to each other (see Figure 1). The frequency-controlled light sources perform constant light–dark adjustment, which is necessary for daylight-independent color measurement. The combination of the spectral values of the incident light, weighted with the functions of the interference filters applied to the photodiode, results in the output currents of the color sensor ICs. The resulting normalization of the measured values results in the absolute XYZ standard spectral values [31,32]. The system is supplied with a 24 V DC voltage source.

Light-emitting diodes with a color temperature of 5700 K (daylight white) provide active illumination of the measured objects for the acquisition of the true color values. This sensor is sensitive in the IR channel to a wavelength λ = 850 nm. This results in four output variables per sensor with the tristimulus values X, Y, Z, and IR = 850 nm. With a housing width of 492 mm, the detection range of the array is 500 mm, which corresponds to the standardized nozzle distance for field sprayers. In practical applications, an array can control a single nozzle valve. A master array can be connected with up to four slaves and transmit the recorded reflection data to a computer.

Software developed at the Institute of Agricultural Engineering is used to convert the reflectance values from the tristimulus values X, Y, and Z into the CIELab color space; this is saved as a csv file together with the recorded IR channel. In the color system developed in 1931 by the Commission Internationale de l′Éclairage (CIE), the color space is spanned by a three-dimensional coordinate system. The values of the luminance L channel (brightness) are displayed on the z-axis with values between 0 (black) and 100 (diffuse white). The values of the a channel are displayed on the x-axis with a range of −128 (green) to 127 (red). The values of the b channel are displayed on the y-axis with the range of −128 (blue) to 127 (yellow) [33]. In contrast to the tristimulus values X, Y, and Z, the CIELab color space differs by considering the brightness; thus, it is more similar to the color perception of the human eye [34]. The influence of ambient light should be eliminated to ensure the reproducibility of field measurements. Fluctuations in brightness result from the characteristic leaf positions of the plants, and they are reflected by the L value. Thus, the color values in the a and b channels are recorded independently of the ambient light and represent the true color values of the test objects to be examined [30].

### 2.3. Design of the Field Trial

The experimental setup was equivalent to a block plantation and comprised four plots with different nitrogen fertilization levels (see Figure 2).

The experimental field had a total area of 468 m² and homogeneous soil conditions. The plots had a length of 15 m and a width of 6 m; they were separated by a distance of 6 m and were laid out next to a tramline to allow field traffic. Plot I represented the control unit without a nitrogen supply. In Plot II, standard nitrogen fertilization was applied after the nitrogen demand was determined. The winter barley received 160 kg N ha^−1^, which corresponded to the standard farming practice. Plot III created a shortage in the nitrogen supply by reducing the application by 50% to 80 kg N ha^−1^. Plot IV considered a 50% increase in the nitrogen supply to 240 kg N ha^−1^. The first nitrogen application for tillering regulation was in mid-March after the vegetation dormancy. At the beginning of April, the second dose followed the elongation growth of the plants. Finally, the third application was conducted at the end of April during the grain-filling phase (i.e., quality fertilization). The measuring dates were at the growth stages ‘Biologische Bundesanstalt, Bundessortenamrt und Chemische Industrie’ (BBCH) 32 (stem elongation with node 2 at least 2 cm above node 1), BBCH 39 (flag leaf stage), and BBCH 59 (end of heading) [35]. These growth stages are essential for nutrient supply on demand of winter barley in European cultivation methods. At these growth stages, nitrogen fertilization is applied on demand [36].

The hatched areas in Figure 2 shows where the TrueColor sensors were deployed. Six arrays with a total of 30 sensor units, each with a measurement range of 500 mm, were mounted side by side on an aluminum profile rail, as shown in Figure 3.

The entire working width was 3 m, which corresponded to a standard boom section, and the total measurement area was 45 m² per plot. The measurement area was not changed throughout the test period to facilitate the comparability of the results. The distance from the sensor to the canopy was typically 50 cm, regarding the typical nozzle distance at field sprayers in European cultivation systems. Crossings were conducted at a constant speed of 0.1 m s^−1^, and four replicates were obtained. The remaining 45 m² of the plot (dotted areas in Figure 2) were used to assess the following reference values.

### 2.4. Reference Measurements

In order to determine the degree of plant coverage, 10 images were taken per plot at a distance of 50 cm from the canopy. The photographs were analyzed with the software ImageJ (Wayne Rasband, version 1.53a, public domain). The reflection properties of the background (soil) were recorded as a reference. The fresh mass was sampled from an area of 0.5 m² with three replicates per plot. The plant height was rated in the three different growth stages for 10 plants per plot. A chlorophyll meter (SPAD-502, (Konica Minolta, Chiyoda, Japan)) was used to determine the chlorophyll content for 20 plants per plot. Measurements were conducted with a single-photon avalanche diode. During the calibration process, the processor converted the current generated by red light ($\lambda_{650}$) and IR radiation ($\lambda_{940}$) into a voltage value. The transmission values in the red ($\lambda_{650}^{\prime }$) and IR ($\lambda_{940}^{\prime }$) wavelength range obtained during the measurement of a leaf were added to the calibration values to derive the SPAD value [37]:(1)$$\mathrm{SPAD}\;\mathrm{value}=\log\frac{\lambda_{940}^{\prime }\cdot \lambda_{650}}{\lambda_{650}^{\prime }\cdot \lambda_{940}}$$

The carbon/nitrogen (C/N) ratio was determined by wet chemical analysis and served as a reference for the actual nitrogen uptake. Samples were taken from the plot area for destructive testing with three replicates. A mixed sample of 40 g fresh mass was cut into approximately 3 cm-long pieces, which were weighed before being dried in a drying oven for 24 h at 60 °C. The SPAD value was considered along with the following agronomic parameters for their correlations with the reflection channels: plant coverage, fresh mass, plant height, and nitrogen level.

### 2.5. Statistical Analysis

Statistical analysis was conducted with the software SPSS Statistics, version 23 (IBM, Armonk, NY, USA). The data were tested by single-factor analysis of variance and a post hoc test according to Scheffé with a significance level of α < 0.05 [38]. This test is suitable for use with databases of different sample sizes. The regression models presented in the Results and Discussion section for the entire study period comprise 48 value pairs: three measurement dates with four nitrogen levels and four passes with the sensor array. The reflectance values were recorded at a log interval of 60 ms. The raw data were used to calculate the arithmetic mean per plot and generate the regression models.

## 3. Results and Discussion

### 3.1. Weather Conditions

The year 2019 was characterized by a pronounced dry period extending from early summer to early autumn, which increased the water deficit from 2018 [39]. Regarding the Sielianinov hydrothermal coefficients (K) in Table 1, the winter barley was sown under very dry conditions on 28 September 2019. Figure 4 shows that the precipitation in September was approximately 30% less than the long-term average (from the years 1956–2014).

Except for January 2020, average to above-average rainfall was observed from October 2019 to the end of March 2020, so the crop had a good water supply at the beginning of vegetation growth. From April on, precipitation was significantly lower than usual; when combined with low water reserves from 2018 and 2019, the appearance of the plants was affected by drought stress. Strong waxy layers and yellow discolored leaf tips could be observed from BBCH 39 onward.

Additionally, the hydrothermal Sielianinov coefficient (K) was calculated according to the following formula.
(2)$$K=\frac{P\cdot 10}{T\cdot L}$$

P is the total precipitation per month, T is the average monthly temperature, and L is the number of days per month. The water availability can be assessed using the scale in Table 1 (below) [40]. This underlines the extremely dry (April) and very dry (May) conditions during the measuring months (Table 1).

### 3.2. C/N Ratio

The nitrogen uptake was determined through wet chemical analysis and according to the C/N ratio. This served as a basis for the following interpretation of the results with the TrueColor sensor array. Figure 5 displays the C/N ratios of different nitrogen levels versus the vegetation from the stem elongation stage (BBCH 32) to full heading (BBCH 59). All nitrogen levels significantly differed in the C/N ratio within a growth stage.

A negative continuous relation was observed between the nitrogen level and the C/N ratio. Across the growth stages, the carbon content increased more than the nitrogen content. Because of this, the C/N ratio increased with the nitrogen level. An exception was the control unit without fertilization (w/o), for which the C/N ratio decreased from BBCH 32 to BBCH 39. This is because the plants in the control unit were able to absorb mineralized nitrogen (N_min_) from the soil. Figure 5 emphasizes real differences in plant nutrition during the growth stages in question. This analysis is helpful in perceiving significant differences in the C/N ratio and serves as a reference for the following analyses considering the reflection properties of the TrueColor sensor array.

### 3.3. Determination of the Required Detection Area

The size of the detection area is an essential parameter for data processing because the time interval between two readings defines the processing speed. This is an important point of discussion because the measurement spot size of actual systems partially covers only 5% of the working width [27]. Figure 6 shows the reflection data of the sensor channels versus the different nitrogen levels and detection areas in BBCH 32. A detection area of 0.0187 m² per reading corresponds to considering all collected data at a constant speed of 0.1 m s^−1^. The influence of the detection area on the meaningfulness of the reflectance values was determined by artificially reducing the data basis. First, 1% of the measured values were randomly selected, and the mean values and standard deviations of the reflection channels for different nitrogen levels were calculated. This resulted in a detection area per measurement proportional to 1.87 m². The same applied when 0.1% of the collected data were used. Figure 6 shows that the results hardly differed for measurement areas of 0.0187 and 1.87 m² (i.e., total sample size versus 1% of the data basis).

The arithmetic mean of the reflectance values within a nitrogen level differed by an order of magnitude of 1%. With a measurement area of 18.7 m² (i.e., a sample size of 0.1%), however, the reflectance values differed. The values of the a and IR channels for different fertilizer levels showed that a consistent relationship no longer existed between the amount of fertilizer and reflectance value at a detection area of 18.7 m². Thus, a detection area of up to 1.87 m² ensures sufficient measurement accuracy. Increasing the time between two measurements (log interval) at a constant speed reduces the computational requirements, whereas increasing the speed at constant log intervals until this detection area is achieved does not degrade the quality of the data.

### 3.4. Influence of the Plant Coverage on the Measurement Quality

Apart from the data processing speed, the need to process the reflection signals is another essential consideration for the practical application of the sensor array. Figure 7 shows the R² values between the degree of plant coverage and sensor channels for different growth stages (BBCH 32–BBCH 59) and the entire study period (Total). The recorded raw data were used as a basis for the reflection of the mixed area (plant + soil). Using the reflectance values without signal post processing reduces the computational time and resources required. For the data series “Plant Reflection” in Figure 7, the background reflection was eliminated by post processing. The raw and post-processed reflection data were analyzed to determine the impact of background reflection on the correlations of the reflection channels. Except for the post-processed data in BBCH 59, the IR channel showed the lowest correlations by far, and it was not considered any further concerning the plant coverage. With λ = 850 nm, the wavelength range is not appropriate to the reflection properties of photosystems I and II [5]. This explains the lower correlations in contrast to the other reflection channels. The L channel delivered uniformly larger R² values with the post-processed data than with the raw data. This was due to the influence of leaf positions on the brightness of reflected light [30].

The a channel covers the green/red range, which includes the light reflected by plants. Superimposing the b channel (blue/yellow range) accounts for the absorption properties of the pigments [6]. The mixed color brown is of particular importance because it reflects the soil. Hence, the plant coverage demonstrated strong correlations with the a and b channels. For the a channel, signal post processing had a negative impact on R², especially for BBCH 59 and Total. By contrast, the b channel had a consistently high R² with plant coverage. Raw and post-processed data had almost no impact on R². The a and b channels achieved R² = 0.72–0.91 with post-processed data. Thus, these two reflection channels were chosen for detailed consideration. Figure 8 outlines the linear regression models for the a and b channels versus the plant coverage in all growth stages (Total).

Compared with the a channel (R² = 0.715), the b channel had a slightly higher correlation with the plant coverage (R² = 0.731). The gradient of the trend line indicates that the values of the a channel were in a smaller range than those of the b channel regarding plant coverage. Increasing the size of the dataset makes the signal more robust against changing measurement conditions, which is advantageous for calibrating the system. In summary, the results indicated that the b channel had the strongest correlation with the degree of plant coverage for all growth stages. Separate consideration of the background (i.e., signal post processing) is not required.

### 3.5. Correlations of the Reflection Channels to the Agronomic Parameters and SPAD Value

In addition to external test conditions, such as the required detection area and the influence of plant coverage, the correlations of the reflection channels with the agronomic parameters were considered. As shown in Figure 9, the R² value was used as an indicator of the sensitivity of the reflection channels to the plant coverage, fresh mass, plant height, nitrogen level, and SPAD value for different growth stages.

The nitrogen level showed the strongest correlation with the b channel (R² = 0.76) and a slightly weaker correlation with the a channel (R² = 0.59). These reflection channels also demonstrated weak correlations with the plant height (R² = 0.35 and 0.25 for the b and a channels, respectively), so they can be applied to determining nitrogen supply. These results support the findings of [8], who found that the nitrogen level, plant coverage, and SPAD data are highly indicative of the predicted yield. The fresh mass is a strong indicator of plant growth and showed a strong correlation with the L channel (R² = 0.76), which, together with the IR channel, also had a strong correlation with the plant height (R² = 0.72). Linear regression models of the fresh mass and plant height are shown separately in Figure 10 (Section 3.5.1) and Figure 11 (Section 3.5.2).

The weak correlation between the plant height and the a and b channels is beneficial for practical field application. Differences in plant height can be recorded and offset by correlation with the IR and L channels without reducing the informative value of the a and b channels on the nitrogen level.

When the R² values of the agronomic parameters were compared for the entire test period and individual growth stages, a noticeable discrepancy was observed that should be considered in detail. An evaluation scheme was developed to elucidate the suitability of reflection channels to the agronomic parameters for different growth stages on a scale from 1 (very well suited) to 4 (not suited). Table 2 presents the evaluation matrix with grades.

The correlations of the reflection channels with the SPAD values are listed as a reference for an already established optical measurement method. The results indicated that the L, a, and b channels had strong correlations with the SPAD values. In contrast to the R² value for the IR channel and plant height in Figure 9, the IR channel achieved conspicuously poor grades for all agronomic parameters in the individual BBCH stages (Table 2). The IR channel did not exceed R² = 0.5 (grade 4) for all investigated growth stages and agronomic parameters. By contrast, the L channel correlated very well with some of the agronomic parameters.

At the beginning of elongation growth (BBCH 32), the L, a, and b channels showed a strong correlation with the plant coverage. Because nitrogen fertilization up to this stage influences tillering and the number of ear-bearing stems, these reflection channels are highly suitable for monitoring the first fertilization. The a and b channels showed strong correlations with the nitrogen level, except in BBCH 59. The fresh mass has a strong correlation with plant health and yield [19,26]. The grades in Table 2 indicate that the a and b channels are promising for yield-oriented fertilization during the entire fertilization period. The correlation between the fresh mass and reflection channels is further discussed below.

#### 3.5.1. Fresh Mass

Ref. [25] observed a correlation r = 0.72 (R² = 0.52) between the red wavelength range and fresh mass. By contrast, Table 2 indicates much higher correlations of the L, a, and b channels with the fresh mass yield for the different growth stages. This indicates a higher sensitivity to the measured variable and thus a finer gradation concerning possible application methods. In combination with preliminary considerations (i.e., the strong correlation with plant coverage and robustness against plant height), the b channel was selected for further consideration. Figure 10 displays the regression model of the b channel and fresh mass yield for different growth stages.

The value pairs of the point cloud comprise the averaged values of the fresh mass yield and b channel for each plot during the growth stages. Samples were taken from an area of 0.5 m² with three replicates per plot; the fresh mass yields ranged from 1335 g m^−2^ (BBCH 32, without nitrogen supply) to 3654 g m^−2^ (BBCH 59, 150% nitrogen supply). The reflectance values of the b channel were between 10.82 and 17.24. The negative and significant linear correlations (*p* < 0.05) are an excellent basis for calibrating the sensor to the fresh mass yield. This is required to adjust the array to site-specific field conditions and can be applied according to threshold values. These thresholds are caused by the nutrient absorption capacity of the plants [23]. This reinforces the strategy of this study to base site-specific fertilization on threshold values with low requirements of computing power. Because the focus was on determining the nitrogen supply while minimizing the ambient impact, the influence of the plant height was also considered, as discussed below.

#### 3.5.2. Plant Height

The results indicated that the plant height does not affect the reflectance values of the b channel (R² = 0.353). This increases the value of the channel for determining the nitrogen level without environmental influences, and it can be used to compensate for a varying distance between the sensor and canopy (e.g., because of vertical movement of the sprayer boom). The plant height can be correlated with other channels of the TrueColor sensor array. The IR and L channels have similar dimensions and represent the plant height with approximately the same quality (see Figure 11): R² = 0.723 and R² = 0.725, respectively.

The regression lines show almost identical negative slopes that differ only by an offset. These results indicate that the IR and L channels are equally suitable for estimating the plant height. However, the conclusions change if the previously presented findings are considered. The influence of the plant cover (Figure 8) showed that the values of the L channel fluctuate less when reflecting plant and mixed areas; thus, the L channel is more robust against the influence of soil (Figure 7). The evaluation matrix (Table 2) showed that the L channel had a better correlation with the plant height than the IR channel, independent of the growth stage. This again gives evidence for the need to calibrate the sensor array under changing conditions (growth stage, soil conditions, and target fertilization level) to ensure precise fertilizer application. With the L channel, a second fertilization adapted to the plant height can be applied at the beginning of elongation growth. This is robust against changing crop populations if the sensor array is calibrated to a specific site. With R² > 0.7 (Figure 11), demand-controlled fertilizer application at the beginning of the elongation growth stage is clearly possible.

#### 3.5.3. Nitrogen Level

The previous results showed that the b channel has the strongest correlation with the nitrogen level as well as the most robustness against environmental impacts for all growth stages. Figure 12 shows the relationship between the nitrogen level and the b channel.

The table in the figure illustrates the mean reflectance values with corresponding homogeneous subgroups and their amount of data. For each nitrogen level, the arithmetic mean and standard deviation were calculated for all growth stages. The reflectance values for the four nitrogen levels differ significantly, which is illustrated by the different subgroups. With R² = 0.906, the homogeneous subgroups showed a very good fit according to the evaluation scheme in Table 2. In addition to the results for the plant cover, the b channel showed good to very good results for the nitrogen level independent of the background and growth stage. Thus, the b channel is the most suitable among all reflection channels for determining the nitrogen level under the experimental conditions in this study.

## 4. Conclusions

The reflectance values of the CIELab color space and IR channel showed significant differences between applied fertilizer levels in different growth stages. The values of the L, a, and b channels decreased with increasing fertilization, which is in contrast to the results for the IR channel. The values of the IR channel were lowest at the maximum nitrogen level and highest without fertilizer application. Meanwhile, the values of the IR channel were higher at the 50% nitrogen level than at the 100% nitrogen level. This non-monotonic course and consistently lower R² with the agronomic parameters indicate that the IR channel should not be used to control nitrogen fertilization. The sensor array should be calibrated against an unknown background (soil) and crop (growth stage, genotype) for demand-controlled nitrogen supply. This is much easier if there is a strong relationship between the nitrogen level and reflectance value, which is observed for the L, a, and b channels. Additionally, the data basis for all reflection channels can be reduced, so the measurement area can be enlarged without affecting the signal quality. The hypothesis that the TrueColor sensor array can achieve good results when measuring the plant coverage was confirmed. Additionally, a strong correlation was observed between the fresh mass yield and the a and b channels. Moreover, the b channel was robust against fluctuations in the plant coverage and plant height. In contrast, the L channel did not perform very well in the advanced growth stages.

Overall, the TrueColor sensor array is suitable for determining the nitrogen supply. Current results are limited on winter barley and should be extended. Good field performance can be achieved with a large working width when the sensor array is cascaded. The fast signal processing and high spatial resolution facilitate the application of this sensor to precision agriculture. Furthermore, the use of low-cost components and increased yield due to more plant-specific fertilizer supply at low yield levels improve the cost-effectiveness of this sensor application. The next development steps include implementing the sensor array in a standard field sprayer for liquid fertilizer application in an online process. The sensor array will also be tested for other types of crops and in perennial field experiments.

## Figures and Tables

**Figure 1 sensors-22-06032-f001:**
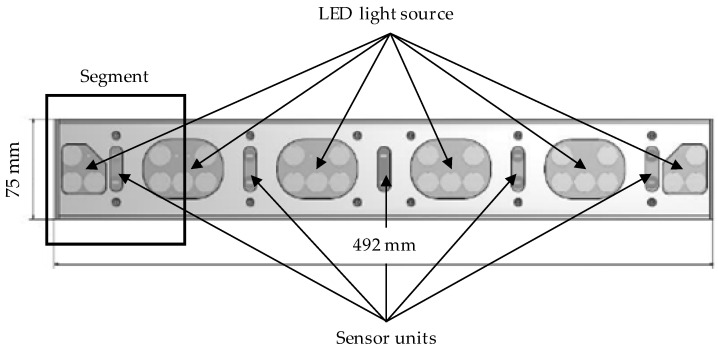
Design and dimensions of the TrueColor sensor array.

**Figure 2 sensors-22-06032-f002:**
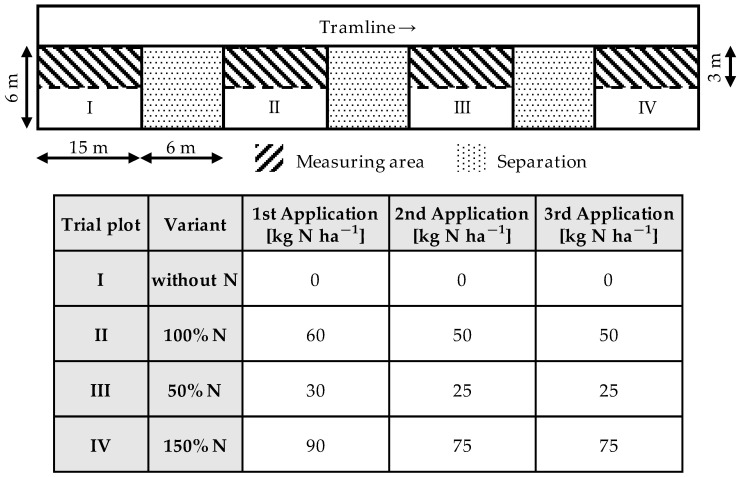
Experimental setup (**top**) and fertilizer rates per application (**down**).

**Figure 3 sensors-22-06032-f003:**
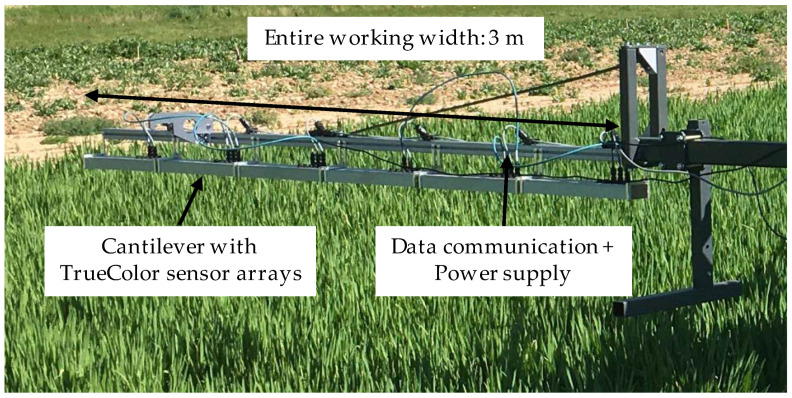
Experimental setup with extension arm for TrueColor sensor arrays.

**Figure 4 sensors-22-06032-f004:**
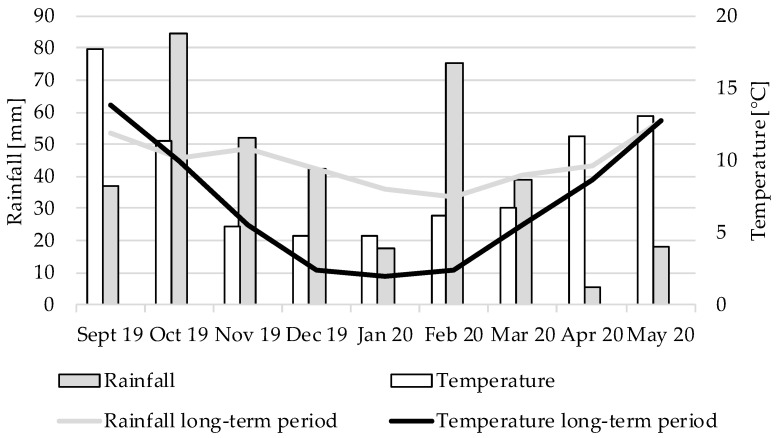
Monthly precipitation quantities and temperature over the 2019/2020 test period in comparison to the long-term mean precipitation and temperature at Campus Klein-Altendorf.

**Figure 5 sensors-22-06032-f005:**
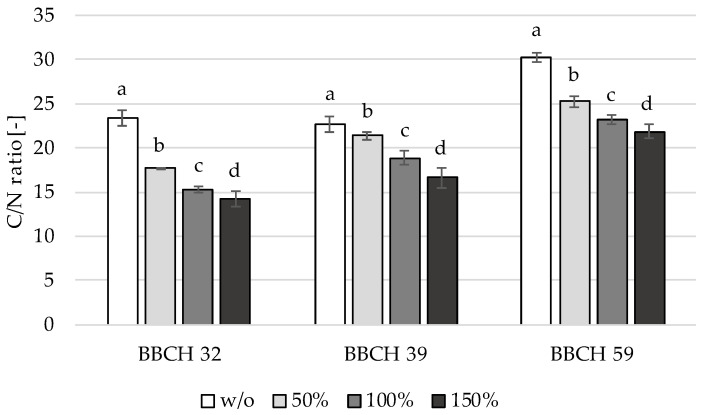
Development of C/N ratio with significance levels a–d for winter barley in BBCH 32 (*p* = 2.19782 × 10^−8^), BBCH 39 (*p* = 1.0403 × 10^−17^), and BBCH 59 (*p* = 1.64079 × 10^−10^) with different nitrogen levels.

**Figure 6 sensors-22-06032-f006:**
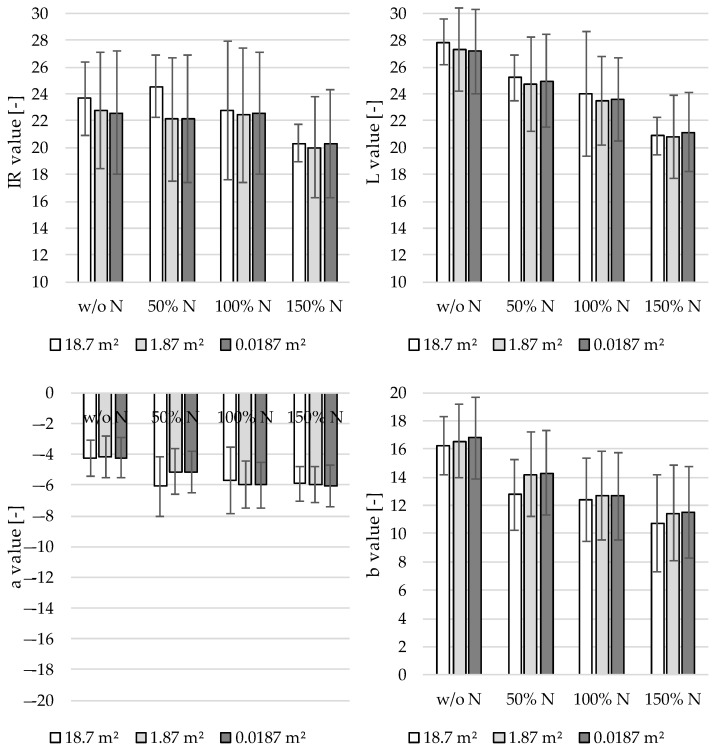
Reflection values of the sensor channels for BBCH 32 with variation of the detection areas.

**Figure 7 sensors-22-06032-f007:**
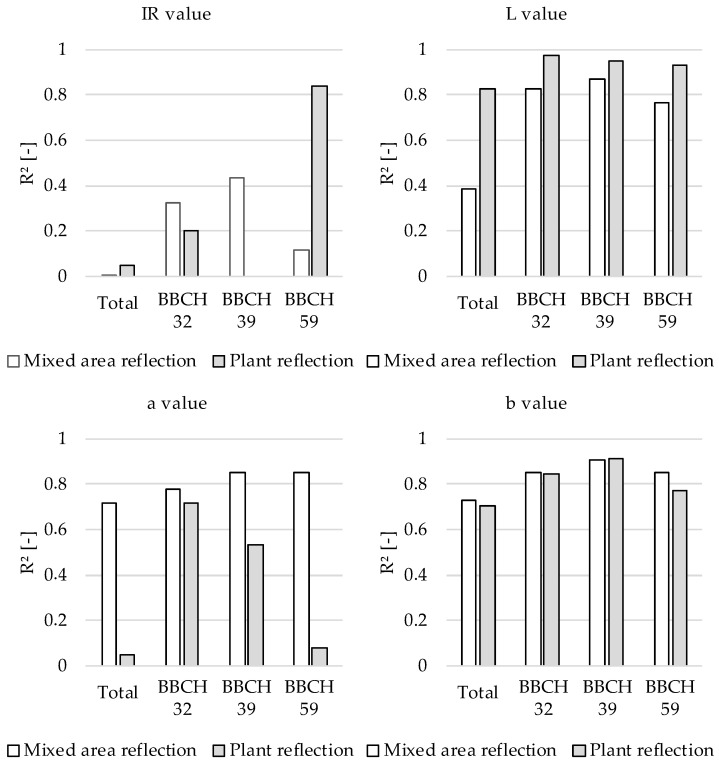
Coefficients of determination (R²) between plant coverage and reflection channels of the TrueColor sensor.

**Figure 8 sensors-22-06032-f008:**
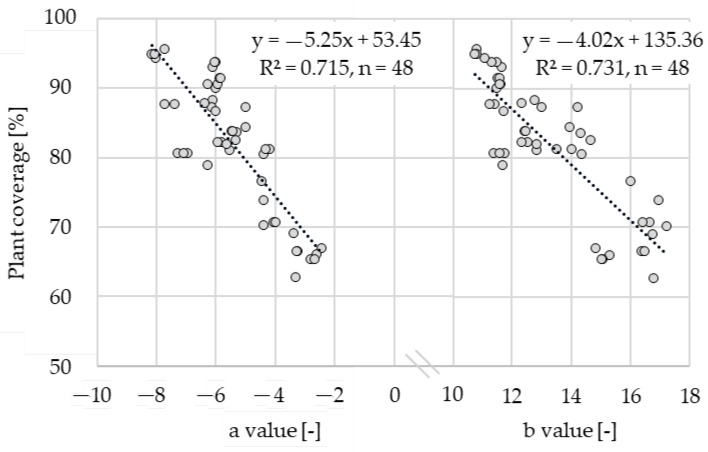
Linear regression models of plant coverage with mixed area reflectance values of a (*p* = 4.0003 × 10^−14^) and b channel (*p* = 1.0449 × 10^−14^) in the growth stages BBCH 32, 39, and 59.

**Figure 9 sensors-22-06032-f009:**
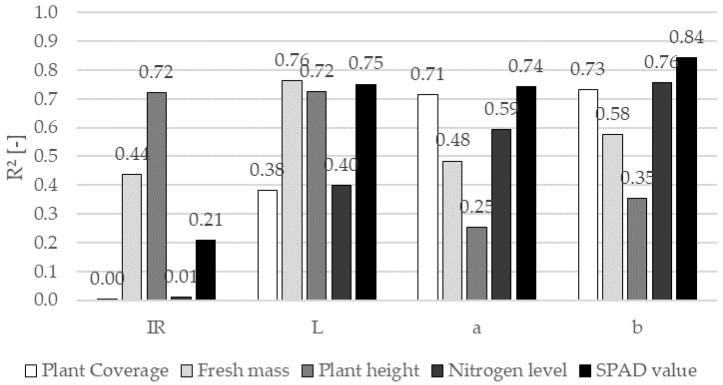
R² of the reflection channels with respect to plant coverage, fresh mass, plant height, nitrogen level and SPAD value, accumulated for the inquired growth stages.

**Figure 10 sensors-22-06032-f010:**
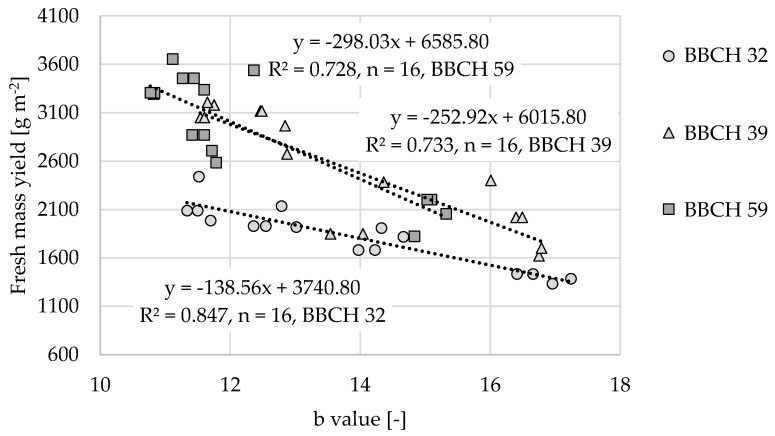
Linear regression models of the fresh mass versus reflection values of the b values for the growth stages BBCH 32 (*p* = 2.259 × 10^−20^), BBCH 39 (*p* = 8.455 × 10^−15^), and BBCH 59 (*p* = 1.322 × 10^−14^).

**Figure 11 sensors-22-06032-f011:**
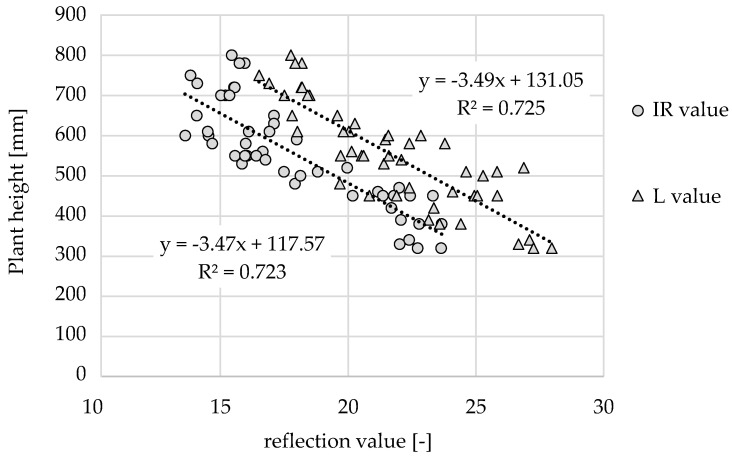
Linear regression model of the plant height with the reflection values of the IR (*p* = 2.0753 × 10^−14^) and L channel (*p* = 1.7399 × 10^−14^) over the growth stages BBCHC 32, 39, and 59.

**Figure 12 sensors-22-06032-f012:**
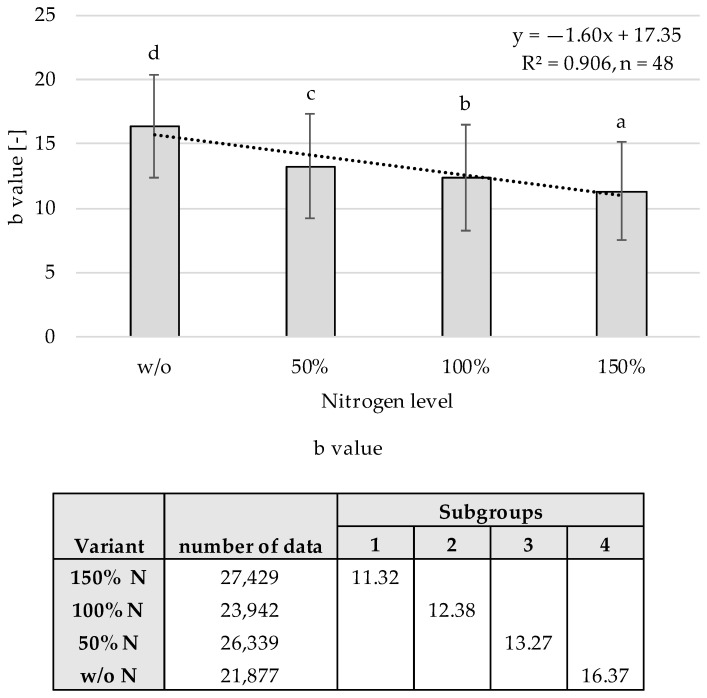
Statistical evaluation of the correlation between b channel and Nitrogen levels with significance levels a–d for the growth stages BBCH 32, 39, and 59 regarding a post hoc test according to Scheffé.

**Table 1 sensors-22-06032-t001:** Hydrothermal Sielianinov coefficients (K) during the growing season according to the Meteorological Station at Campus Klein-Altendorf.

	Sept 19	Oct 19	Nov 19	Dec 19	Jan 20	Feb 20	Mar 20	Apr 20	May 20
K	0.69	2.39	3.22	2.85	1.18	4.33	1.88	0.15	0.45

Extremely dry, K ≤ 0.4; very dry, 0.4 < K ≤ 0.7; dry 0.7, < K ≤ 1.0; quite dry, 1.0 < K ≤ 1.3; optimal, 1.3 < K ≤ 1.6; quite humid, 1.6 < K ≤ 2.0; humid, 2.0 < K ≤ 2.5; very humid, 2.5 < K ≤ 3.0; extremely humid, K > 3.0.

**Table 2 sensors-22-06032-t002:** Evaluation matrix of the reflection channels concerning their specific sensitivity for the recorded agronomic parameters and SPAD value.

		IR	L	a	b	Grade	Suitable	>R²
BBCH 32	Plant coverage	4	1	2	1	1	very good	0.8
	Fresh mass	4	2	1	1	2	good	0.7
	Plant height	4	3	4	3	3	partially	0.5
	Nitrogen level	4	1	1	1	4	not	0
	SPAD value	4	1	1	1			
BBCH 39	Plant coverage	4	1	1	1			
	Fresh mass	4	3	1	2			
	Plant height	4	3	2	2			
	Nitrogen level	4	2	1	1			
	SPAD value	4	2	1	1			
BBCH 59	Plant coverage	4	2	1	1			
	Fresh mass	4	2	2	2			
	Plant height	4	3	3	3			
	Nitrogen level	4	3	2	2			
	SPAD value	4	1	1	1			

## Data Availability

Data and intellectual property belong to the University of Bonn; any sharing needs to be evaluated and approved by the University.

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
