# Peer review of "Capability of the TrueColor Sensor Array for Determining the Nitrogen Supply in Winter Barley (Hordeum vulgare L.)"

_sensors, 2022, doi:10.3390/s22166032_

Round 1

Reviewer 1 Report

The manuscript presents an evaluation study of a new mobile sensor for crop nitrogen monitoring, using barley as an example. The evaluation was carried out in a near real-life environment. The manuscript presents the result of the original research, which is presented in an interesting way.

The introduction provides the necessary information to give a background to the research. It has been prepared correctly, however, with a certain focus only on European, especially German, research. The introduction of a broader view of agricultural mobile vegetation/phenological state sensors would have enriched the manuscript.

The methodology of the procedure, the results of the work and their discussion and conclusions are described in a clear manner.

The manuscript contains several minor shortcomings, the removal of which will allow it to be published.

Specific comments and suggestions:

L27: I suggest removing from keywords „ Variable Rate Technology”. The paper does not contain research in this area and the keyword " Variable Rate Technology" only appears in the keywords. I also put the validity of the other keywords under consideration.

L31: Sentence „Nitrogen from crop fertilizer that is not absorbed by plants has negative effects on the environment, such as groundwater pollution.” requires a reference.

L86: “17% for the Yara N-Sensor ALS, 11% for GreenSeeker, 5.5% for Fritzmeier ISARIA, and 5% for AgLeader OptRx [26].”. Please provide the correct product name and origin/manufacturer. The names given are no longer valid, e.g. Fritzmeier ISARIA already has a different name.

L156: Figure 2. 1. Application; 2. Application; 3. Application.  Change to: 1st application, 2nd..

L168: Please provide references for the BBCH scale. Sentence „The measuring dates were at the growth stages ‘Biologische Bundesanstalt, Bundessortenamrt und Chemische Industrie’ (BBCH) 32, BBCH 39 and BBCH 59” needs rewording. A verbal description of the BBCH phases would facilitate readability. They are in the text, but it is worth combining them. It is worth explaining why these particular phases were chosen. This is likely to be standard in barley agronomy in the region.

L182: software ImageJ (public domain). Please specify manufacturer and software version.

L187: „SPAD-502, (Konica Minolta, Chiyoda, Japan))”  For other commercial technical products, the manufacturer's city was not stated. I propose to remove this exception.

L201: SPSS Statistics (IBM, Armonk, USA). As above.

L202: “The data were tested by single-factor analysis of variance and a post hoc test according to Scheffé” .  A necessary reference for Scheffé.

L210: 3.1. Weather conditions. In the description of the climatic conditions, only the precipitation parameter is given. For a complete description of the conditions, it would be advisable to give the full pluviometric conditions hence the addition of temperature, also with reference to multi-year, seems advisable. The introduction of the Sielianiow hydrothermal coefficient could be considered.

L192: Equation 1 has not been quoted. Similarly, other equations.

L414: Figure 12 needs to be corrected „ Scheffé”? „150 %” >> „150%”

L461: „Funding: This research was funded by the Deutsche Bundesstiftung Umwelt (DBU) and the German Federal Ministry of Education and Research (BMBF), grant numbers Az 31602-05 and Az 33340.” I suggest choosing one language version of the institution, either in German both or in English.

Author Response

Dear Reviewer 1,

Kind Regards,
Andreas Christ.

Reviewer 2 Report

The authors developed a sensor for detecting nitrogen supply. I have the following questions and suggestions for improving the manuscript.

Line125:”… perform constant light–dark adjustment, which is necessary for daylight-independent color measurement.” The Truecolor sensor’s wavelength range is from 380 – 1000 nm, which covers the visible-NIR range. The main challenge is the sensor’s interference between sunlight and light sources.

Fig.6 : please give more explanation about the large error bar and how to minimize the error bar.

Fig.12: I cannot understand “Scheffé” here. Please check the academic writing.

The authors used six arrays with 30 sensor units in the proposed system. How about the overall cost?

Line 178: why did you set 50 cm from sensors to canopy? Is it possible to set further distance with fewer sensors?

Please unify the format of figures.

Author Response

Dear Reviewer 2,

Kind Regards,
Andreas Christ.
